# Prediction Method of Steel Corrosion Rate Based on the Helix Distributed Sensor

**DOI:** 10.3390/mi13111868

**Published:** 2022-10-30

**Authors:** Jian-Zhi Li, Yi-Yao Zhao, Jun-Jie Wang

**Affiliations:** 1Key Laboratory of Structural Health Monitoring and Control, Shijiazhuang Tiedao University, No. 17, Beierhuan East Road, Changan District, Shijiazhuang 050043, China; 2School of Materials Science and Engineering, Shijiazhuang Tiedao University, No. 17, Beierhuan East Road, Changan District, Shijiazhuang 050043, China

**Keywords:** steel corrosion, spiral distributed sensor, reinforced concrete

## Abstract

Corrosion of steel bars is of great significance for safety and service life of reinforced concrete structures. This work develops a prediction method for steel corrosion mass loss rate before the crack of concrete structure based on a spiral distributed fiber optic sensor. Reinforced concrete sample instrumented with a spiral distributed fiber optic sensor were prepared. The mathematic relationship between the corrosion mass loss rate of steel bar and the spiral distributed strain is theoretically derived. Meanwhile, numerical analysis by MATLAB shows that these parameters such as the protective layer thickness, corrosion mass loss rate, bar diameter, corrosion expansion coefficient have a remarkable influence on spiral distributed strain. Additionally, electrical accelerated corrosion experiment was performed on the reinforced concrete specimens. The helix strain along the distributed sensor was used to evaluate the corrosion mass loss of steel bar. Further, the influencing factors on the corrosion sensitivity are illustrated here and the corrosion mass loss rate before concrete crack is also quantified. This research provides insights into the corrosion deteriorate mechanism.

## 1. Introduction

Many engineering structures are made of reinforced concrete. Steel bars corrosion has been a main challenge for durability of reinforced concrete structures, which has attracted a wide attention of scholars both at home and abroad [1,2,3]. Steel bar corrosion not only re-duces the load-carrying capacity owing to the reduced cross section of the bar, but also degrades the concrete-steel interface and consequently causes concrete cracks which in turn accelerate steel corrosion. It is significant to predict steel bar corrosion of reinforced concrete to ensure its safety and effective asset management. The researchers mainly focused on two aspects, i.e., theoretical mechanics analysis and test methods. Theoretical analysis of corrosion-induced damage of reinforced concrete structure [4,5,6,7,8,9,10] has shown that its degradation process is divided into three stages: corrosion expansion stage, protective layer cracking stage, crack extension stage. Moreover, the reinforcing bar corrosion cannot be effectively evaluated [7,9] before the crack propagates to the surface of the concrete structure. These corrosive cracking models are established through analysis of elastic mechanics. These literatures show that the concrete crack play a significant role for a structural degradation. In addition, the efforts to find some practicable methods for predicting the concrete corrosion have been made [11,12,13,14]. In recent years, due to its small volume, light weight, resistance to electromagnetic interference, stability under chemical attack and long-term monitoring, fiber optic sensors have a remarkable development. The comparison with traditional corrosion monitoring methods is shown in the Table 1. Among them, special FBG sensors [15,16] were used to monitor the corrosion based on the principle of expansion of steel corrosion. An alternative approach is to electroplate a Fe-C film caused strain changes in the FBG and LPFG sensor [17,18,19,20], and thus corrosion was monitored by the central frequency shift of FBG. Another paradigm of FBG corrosion sensor is based on thermal effect [21]. Due to the water content and bond condition between the sensor and steel bar, it is difficult to interpret the sensor data in real application. Moreover, residual strain sensors [11] based on the mechanism of prestress loss were developed to solve the corrosion problem.

Despite of the exciting advancement, the FBG and LPFG sensor only measures monitoring corrosion at a single spot installing a sensor, and the degradation of a corrosion sensor is quick and irreversible [18,19,20,22,23]. To monitor the corrosion condition of a real-scale civil engineering, a large quantity of sensors is necessary to obtain detailed information of different locations. However, due to the space and time randomness of corrosion and the concealment of steel bar, the researchers turned to a fully distributed optical fiber sensor to predict the steel corrosion. fiber optic rings [12,24,25,26] has been developed. Nevertheless, the tightly spaced optical fibers cause a corrosion resistance and degraded bond strength. Additionally, this proposed ring sensor was still a single spot sensor and further was limited to a meter order instead of pinpointing the corrosion sites. To develop a real-time, in-situ corrosion monitoring method and monitor the deterioration process of reinforced concrete, a helix distributed sensor [27,28,29,30,31,32,33,34,35] has been developed based on the measurement of expansive stain induced by steel bar corrosion. However, these authors still employed a ring sensor mathematic model to calculated fiber strain rather than a helix pattern model, causing a greater error. Simultaneously, these researches still failed to provide a prediction model for corrosion mass loss.

Thus, this work has two main objectives: (1) deduces a theoretical mathematic model between the corrosion rate of steel bar and optical fiber strain based on a helix pattern rather than a ring pattern, and (2) attains a practicable model between steel corrosion mass loss rate and spiral distributed strain from a helix distributed sensor. The influencing fac-tors on the corrosion sensitivity are illustrated by numerical analysis with MATLAB. Meanwhile, the mathematical model between steel corrosion rate and the spiral distributed strain is experimentally verified and illustrated. This work ultimately provides a quantitative method of predict the corrosion mass loss rate before the concrete crack occurs.

## 2. Structure and Principle of the Distributed Sensor

### 2.1. Strain Sensing Principle of Brillouin Distributed Sensor

BOTDA technology is based on stimulated Brillouin scattering effect. The Brillouin frequency shift depend on the fiber strain and temperature. When the temperature and strain change, the Brillouin frequency will shift. Thus the temperature and strain can be calculated from the Brillouin frequency shift [36]:(1)$$v_{B}(\varepsilon ,T)=v_{B}(0,T_{0})+C_{\varepsilon }\varepsilon +C_{T}(T-T_{0})$$
where T represents the ambient temperature, $v_{B}(\varepsilon ,T)$ is the Brillouin frequency shift; $v_{B}(0,T_{0})$ is the Brillouin frequency shift at $T_{0}$ temperature, $C_{\varepsilon }$ and $C_{T}$ represent the sensitivity strain and temperature coefficient, respectively. The strain and temperature sensitivity coefficient were 0.0497 MHZ/με and 1.17 MHZ/°C, respectively. Thus, the fiber strain can be attained: (2)$$\varepsilon =[v_{B}(\varepsilon ,T)-v_{B}(0,T_{0})-C_{T}(T-T_{0})]/C_{\varepsilon }$$

### 2.2. Temperature Compensation Principle of FBG

Due to a temperature change, the wavelength shift of FBG free from stress can be expressed as:(3)$$\frac{\Delta \lambda_{B}}{\lambda_{B}}=\left(\alpha_{s}+\zeta_{s}\right)\Delta T$$
where $\alpha_{s}=\frac{1}{\Lambda }\frac{\Delta \Lambda }{\Delta T}=\frac{1}{L}\frac{\Delta L}{\Delta T}$ represents the thermal expansion coefficient of the fiber, which describes the relationship between the grating period and temperature; $\zeta_{s}=\frac{1}{n_{eff}}\cdot \frac{\Delta n_{eff}}{\Delta T}$ represents the thermal optical coefficient of the fiber, which describes the relationship between the effective refractive index of the grating and temperature. Therefore, the temperature change can be obtained from the wavelength shift. The relationship between bare grating wavelength and temperature can be calculated as:(4)$$\Delta \lambda_{B}=k\Delta T$$
where k is 11 pm/°C. FBG located in the rubber sleeve is free in this experiment due to the much larger sleeve diameter than the diameter of FBG and is influenced by only temperature.

### 2.3. Structure Design of Distributed Sensor

A spiral distributed optical fiber sensor was developed here, as shown in Figure 1. The sensing fiber was wrapped on the steel rebar and the mortar layer with a given winding helix angle. This winding pattern decreased light loss and consequently, lengthen the tested fiber sensor. Compared with a ring sensor, the proposed spiral sensor here has two extinguished characteristics, one is the measurement of corrosion mass loss rate along the steel bar, another is to lengthen the tested fiber sensor. In order to attain the practicability of distributed fiber optic corrosion sensors, a given winding helix angle difficult to measure was transformed into a wrapped interval called a winding pitch. A small interval may reduce the rebar-concrete bond strength and increase optical light loss, while a coarse interval may miss corrosion information and lengthen the tested length. Meantime, a smaller interval will generate noise in the measurement results, and a larger interval will omit corrosion information, which fails to accurately reflect the corrosion of reinforcement. An optimal wound angle is indispensable to consider both bond strength and the length of the distributed corrosion sensor. The corresponding spiral angle is 30° and 60°. Due to an unmeasurable spiral angle, the winding pitch is adopted as a predominant wound parameter. Then, the pitch in this study is correspondingly 8.7 cm and 6.5 cm.

## 3. Prediction Model of Corrosion Rate

### 3.1. Quantification of Corrosion Rate Based on Spiral Fiber Strain

Corrosion rate before concrete cracks occur is discussed based on elastic mechanics theory, while the distributed fiber strain is dependent on the crack width after the concrete crack. Therefore, we make the following assumptions:(1)According to elastic mechanics theory, the fiber model can be simplified to a two-dimensional model, due to the corrosion expansion independent on the steel bar length, and concrete material is further assumed to be isotropic [9,37].(2)The actual corrosion duration and corrosion velocity depends on the electrical current intensity;(3)It is assumed that the steel bar corrosion expansion is uniform, then the thick wall cylinder elastic mechanics model can be used to describe the corrosion process [9,38].

When pristine concrete and steel bar are considered as an integrated object, the corrosion process of reinforcement concrete is divided into three stages, as shown in Figure 2. In the first stage, the corrosion product fills the pore in the contact interface [28,32]; in the second stage, the accumulated corrosion product generates tensile stress and the concrete deformation occurs; in the third stage, the through-crack of reinforcement concrete occurs. Based on literature [7,9,39], the concrete deformation $\delta_{con}$ from corrosion product can be expressed as:(5)$$\delta_{con}=\frac{q}{E_{ef}}[(1+\mu_{c}){\frac{R(R+c)}{2Rc+c^{2}}}^{2}+(1-\mu_{c})\frac{R^{3}}{2Rc+c^{2}}]$$
where q is the tensile stress from the corrosion production, $E_{ef}$ and $\mu_{c}$ are modulus of elasticity (MPa) and Poisson ratio of concrete [40], R is the diameter of steel bar(mm), c is the thickness of concrete cover(mm), $\mu_{c}$ is the Poisson ratio of concrete.

According literature [39], the tensile stress induced by corrosion production before the crack is given:(6)$$q=\frac{[\sqrt{(n-1)\rho +1}-1]R}{\frac{n(1-\mu_{r}^{2})\sqrt{(n-1)\rho +1}R}{E_{r}\left\{\left[\left(1+\mu_{r}\right)n-2\right]+2/\rho \right\}}+\frac{1}{E_{ef}}[(1+\mu_{c}){\frac{R(R+c)}{2Rc+c^{2}}}^{2}+(1-\mu_{c})\frac{R^{3}}{2Rc+c^{2}}]}$$
where Er and μr are the modulus of elasticity (Mpa) and Poisson ratio of the rust, respectively. The value of μr is 0.49. Er is determined as $Er=6000(1-2\mu_{r})$, n is the volume expansion coefficient of rust, typically 2–4.

Plugging Equation (4) into Equation (3), the thickness of the rust $\delta_{con}$ can be calculated as follows:(7)$$\delta_{con}=\frac{\left[\sqrt{(n-1)\rho +1}-1]R\frac{1}{E_{ef}}[(1+\mu_{c}){\frac{R(R+c)}{2Rc+c^{2}}}^{2}+(1-\mu_{c})\frac{R^{3}}{2Rc+c^{2}}\right]}{\frac{n(1-\mu_{r}^{2})\sqrt{(n-1)\rho +1}R}{E_{r}\left\{\left[\left(1+\mu_{r}\right)n-2\right]+2/\rho \right\}}+\frac{1}{E_{ef}}[(1+\mu_{c}){\frac{R(R+c)}{2Rc+c^{2}}}^{2}+(1-\mu_{c})\frac{R^{3}}{2Rc+c^{2}}]}$$

Then the combined diameter of steel bar and the rust before the concrete begin to crack is determined:(8)$$D=d-2\delta +2\delta_{con}$$

As shown in Figure 3, the initial circumference of steel bar is $C_{0}$. When the diameter of rebar is increased to D arising from steel bar corrosion expansion, the circumference of steel bar is correspondingly $C_{1}$. Unwinding optical sensing fiber based on a spiral wound angle, fiber optic strain can be expressed as follows:(9)$$L_{0}=\sqrt{C_{0}^{2}+L^{2}}$$
(10)$$L_{1}=\sqrt{C_{1}^{2}+L^{2}}$$
(11)$$\varepsilon_{f}=\frac{\sqrt{\pi^{2}D^{2}+L^{2}}-\sqrt{\pi^{2}d^{2}+L^{2}}}{\sqrt{\pi^{2}d^{2}+L^{2}}}$$
where L is the winding pitch, $L_{0}$ is the initial length of steel bar, $L_{1}$ is the ultimate length of rebar caused by corrosion expansion; $\varepsilon_{f}$ is the fiber strain caused by corrosion expansion.

Plugging Equation (6) into Equation (9)
(12)$$\varepsilon_{f}=\frac{\sqrt{\pi^{2}{\left[d-2\delta +2\delta_{con}\right]}^{2}+L^{2}}}{\sqrt{\pi^{2}d^{2}+L^{2}}}-1$$

Plugging Equations (5) and (6) into Equation (10), the mathematical relation between fiber strain and corrosion rate is derived:(13)$$\varepsilon_{f}=\left\{\frac{\sqrt{\pi^{2}{\left[d-2P_{vp}d_{p}+2A\right]}^{2}+L^{2}}}{\sqrt{\pi^{2}d^{2}+L^{2}}}-1\right\}\times 10^{6}$$
(14)$$A=\frac{\left[\sqrt{(n-1)\rho +1}-1]R\frac{1}{E_{ef}}[(1+\mu_{c}){\frac{R(R+c)}{2Rc+c^{2}}}^{2}+(1-\mu_{c})\frac{R^{3}}{2Rc+c^{2}}\right]}{\frac{n(1-\mu_{r}^{2})\sqrt{(n-1)\rho +1}R}{E_{r}\left\{\left[\left(1+\mu_{r}\right)n-2\right]+2/\rho \right\}}+\frac{1}{E_{ef}}[(1+\mu_{c}){\frac{R(R+c)}{2Rc+c^{2}}}^{2}+(1-\mu_{c})\frac{R^{3}}{2Rc+c^{2}}]}$$
where the unit of fiber strain is με.

The magnitude of corrosion expansion ratio [41,42] has been investigated. When the main corrosion product is Fe_2_OH, and Fe_3_O_4_ Fe_2_O_3_, the expansion volume of Fe_2_O_3_ is identical with that of Fe_3_O_4_. The volume of Fe_3_O_4_ and Fe_2_O_3_ in corrosion product is accounted for 85%, *n* is approximately from 1.79 to 2.56, typically 2. The elastic modulus of concrete is 35 GPa.

### 3.2. Calculation and Influencing Factors of Fiber Strain

Figure 4 show that the relationship between the protective layer thickness and the helical distributed strain. It decreases rapidly with the protective layer thickness to a steady strain. When the protective layer thickness is greater than 35 mm, the helix strain is unrelated to the protective layer thickness. In other words, the reinforced concrete with a thinner protective layer is easily deformed, then generate a greater helix strain. Additionally, the spiral strain depends on the corrosion mass loss rate and rise to a constant and ultimately keep steady. Figure 5 shows the relations between the corrosion mass loss rates with the helix distributed strain. It increases rapidly to 500 με with the corrosion mass loss rate, and subsequently remain stable. When the protective layer thickness is 67 mm and 142 mm, no significant difference was observed from the distributed strain-mass loss rate curves. Simultaneously, the spiral distributed strain is proximately proportional to the corrosion mass loss rate. The regression error is approximately 50 με, which is acceptable due to a measured error induced by the BOTDA instrument. The corrosion sensitivity is 92.8 με/% beyond 0.5% corrosion mass loss rate, shown as in Figure 5.

Meanwhile, the author analyzed the influence of the steel bar diameter of 5 mm, 10 mm, 16 mm, 20 mm on helix distributed strain (c = 67 mm, *n* = 2, δ = 0.001 mm), as shown in Figure 6. The optical fiber strain firstly increases with the diameter of steel bar, subsequently rise to a peak value and then drop slowly. Therefore, to attain a higher sensitivity, the steel bar diameter is around 25 mm.

The influence of corrosion expansion coefficient on the spiral fiber optic strain is shown in Figure 7. The helical strain is mainly dependent on the corrosion expansion coefficient *n*. As shown in Figure 7, fiber distributed strain with corrosion expansion coefficient of *n* = 4 increases more sharply than that with the corrosion expansion coefficient of *n* = 2 at initial corrosion duration. Finally, the helix fiber strain remains basically unchanged. The reason for the above phenomena is that a greater corrosion expansion coefficient accelerates the filling process of corrosion products and leads to a higher compress of concrete. Thus, the corrosion rate threshold (generating an actual deformation of concrete around the bar) is less than that the specimen with a smaller corrosion expansion coefficient. However, the corrosion expansion coefficient *n* is difficult to be precisely measured, and the helix fiber strain is consequently unable to be calculated by Equation (11).

To sum up, these parameters such as the protective layer thickness, corrosion mass loss rate, bar diameter, corrosion expansion coefficient has a remarkable influence on the spiral distributed strain.

## 4. Experiment and Test

The optical fiber used in the experiment is a tight-jacketed optical cable in 0.9 mm diameter with a high stain transmission efficiency. The mortar cover is produced with ordinary Portland cement, water, fine aggregate and water reducing agent (Shanshufeng Technology, Hubei, China) in a weight proportion of 1.0: 0.45: 0.1: 0.03. The cement used is P.O 42.5. There is a 200 mm-long straight round carbon steel bar in 16 mm diameter used. To induce a uniform surface property, the bar surfaces were sandblasted to remove the mill scale in order. Subsequently, a spiral distributed sensing fiber with a 30° helix angle was wound with an 8.7 cm interval around the steel bar. Meanwhile, Mortar is cast to a cylindrical mould and a 10 mm-thickness mortar layer is covered on the steel bar. Eventually, a spiral distributed sensing fiber with a 60° helical angle was wound with a 6.5 cm pitch and embedded into concrete with a 67 mm and 142 mm concrete cover, demolded after 24 h, and cured for 28 days. The specimens are named as 2-150# and 2-300#, respectively. To the end, the concrete end surfaces were sealed using silica gel to prevent from the outflow of corrosion products.

After 28-days of concrete curing, the concrete specimens were subjected to electrochemically accelerated corrosion to obtain a continual corrosion loss. Before corrosion test, the fabricated concrete samples were immersed in 5% sodium chloride solution for 48 h. The content of admixed chloride in this study far exceeded the critical chloride threshold. Hence depassivation was warranted for the adopted polished surface condition. A copper wire was subsequently soldered at each end of steel bar. Meanwhile, the steel bar and a copper beam were, respectively, connected to the positive and negative electrodes of the power supply. The initial designed corrosion current is 0.1 A constant current. The calculation of steel corrosion rate follows Faraday’s law. Both ends of the fiber jumper are connected to the Pump and Probe end of BOTDA. The FBG temperature sensor is connected to SM130 Demodulation System and placed in the U-shaped slot in a free state. The experimental setup is detailed in Figure 8. The specimens were tested at room temperature and measured the strain distributions along the sensing fiber every 10 min. The moment when the fiber strain holds constant is defined as completion of the electrochemical test.

## 5. Testing Results and Discussion

### 5.1. Temperature Effect in Electric Accelerated Corrosion Test

For the specimen with a 67 mm protective layer (mortar layer), the electrical current generated more heat in the corrosion process. FBG sensor (Tongwei Technology, Beijing, China) was used to measure the temperature around the steel bar. It is observed from Figure 9 that the internal temperature magnitude of reinforced concrete specimen generated by an applied electrical current varied from 0 °C to 4 °C in the process of electric corrosion. The bottom and top horizontal axis represents the accumulated corrosion time and steel bar corrosion loss rate. The vertical axis represents the magnitude of internal temperature variation calculated by the wavelength shift of FBG sensor. The inner temperature rises sharply shortly after accelerated corrosion starts. Subsequently, when through-thickness cracks occurred, there is a fast temperature drop. The phenomena are attributed to ingress of sodium chloride solution through the cracks. Ultimately, the steel bar temperature is identical with the corrosion solution temperature after cracking. For the specimen with a 142 mm cover layer, as shown in Figure 10, the internal temperature magnitude of the reinforcement concrete varies between −4 °C and 16 °C, and the temperature variation is similar with the concrete specimen with a 67 mm protective layer. In addition, the temperature fluctuations are in a good agreement with the applied corrosion current in the corrosion process and ambient temperature after a through crack was generated. After concrete cracks, the temperature fluctuations depend on the ambient room temperature. Then it indicates that the temperature has a remarkable influence on the Brillouin frequency shift, further causing a measured error of helix distributed strain. In a word, temperature should be measured in the duration of electrical corrosion, otherwise the greater test error is generated.

### 5.2. Variation Law of Actual Optical Fiber Strain with Corrosion Time

The strain-time curves before and after temperature compensation for specimens 2-150# and 2-300# are shown as Figure 11, Figure 12 and Figure 13, and the test parameters for significance of fitting curve is list in Table 2, Table 3, Table 4, Table 5, Table 6 and Table 7. The vertical axis represents the raw strain measured from the spiral distributed fiber sensor and actual strain subtracted the induced-strain by the temperature fluctuation. The top and below horizonal axes represent the mass loss rate and accumulated corrosion duration. The actual distributed strain of sensing fiber represents the corrosion-induced strain, whereas raw strain comprises that induced not only by corrosion expansion but also a varied temperature due to a corrosion current.

(1)Reinforced concrete specimen with a 67 mm concrete cover

The strain-time curves are shown in Figure 11 before and after temperature compensation of 2-150# specimen. Since the distributed sensor located in steel bar layer is broken during the mold removal process, only the distributed sensor wrapped on the mortar layer is available. It is observed from the figures that the strain measured from the distributed sensor gradually grow, subsequently increases rapidly after 50 h. To the end, the strain increases to a maximum value and remain constant. Additionally, the concrete specimen cracks at 50 h, close to the observed crack time (55 h), the difference between the predicted cracking corrosion rate and the actual corrosion rate, observed crack time of 55 h, is approximately 0.17%. Enlarging the curve before specimen cracking, the raw strain along the distributed sensor increases to 150 με sharply before 2 h in the inner illustration. Subsequently, the fiber strain increases linearly. Meanwhile, a rapid strain growth in first 2 h induced by the current is consistent with the temperature variation (Figure 9). Thus, it is essential to use a temperature sensor for decreasing a tested strain error. In addition, the actual optical fiber strain (0~11 h) remains constant, which is defined as the corrosion rate threshold (0.36%), being consistent with the existed literature [28,32]. The thickness of porous zone δ is approximately 0–20 um [23], which contributes to the corrosion rate threshold. This reason is that the corrosion product used to fill porous area offsets the corrosive force, simultaneously alleviates the strain from a distributed sensor. Likewise, as more rusts are produced and accumulated at the concrete-steel interface, the expansive strain begins to increase until the corrosion products generate a compress on the surrounding concrete. Consequently, more corrosion products generate an expansive strain causing a distributed fiber deformation. Then the corrosion rate threshold is in turn increased.

Meanwhile, the raw and actual strains in the distributed sensor monotonically increase over time and corrosion mass loss rate. Each curve shows four stages, which the I, II, III stages are consistent with the degradation of reinforcement concrete in Figure 2. Regression analysis is performed to determine the equation of stages 2, 3 and 4. Before reinforcement concrete cracks, the slope of the fitting equation between the distributed strain and corrosion mass loss rate is 125 με/% as shown in Table 2, while the slope of the fitting curve between the distributed strain and corrosive time is 3.37 με/h as shown in Table 3, less than 16.8 με/h in this literature[28,32]. Possibly, their difference is attributed to their distinct thickness of concrete cover. This 3.1 section shows that the helix strain is dependent on the protective layer thickness greater than 35 mm. Simultaneously, the *P* value of regression equation less than 0.05 illustrates that the linear arrogation equation developed in this work is credible.

(2)Reinforced concrete specimen with a 142 mm concrete cover

Figure 12a and Figure 13a indicates the relationship between corrosion time and the distributed strains from the distributed sensor located in steel bar layer and mortar layer of specimen 2-300#. At first 2 h, the raw strain wrapped on the steel bar layer and the mortar layer increases rapidly to 200 με and 250 με up to 40 h. Subsequently, the optical fiber strain rises linearly. At the time t = 78 h, the raw strain of optical fiber in the reinforcement layer increases to 1000 με and that in the mortar layer increases to 750 με. At 80 h, the raw strain in the reinforcement layer and mortar layer decreases sharply (the reinforcement layer decreases to 500 με and the mortar layer to 300 με), subsequently, continue to decrease rapidly to 300 με and 150 με at 120 h. The above-mentioned phenomena are consistent with the current fluctuations adjusted from 0.1 A to 0.05 A and 0.03 A. At the time of 225 h, the epoxy resin layer is peeled off. The corrosion experiment is forced to stop, resulting in a strain decrease and then increase (at the marking position in Figure 12a) after continuing to carry on the corrosion experiment. It is also found that the influence of the factors such as current adjustment and sealant leakage on the fiber strain before the crack of concrete are eliminated after temperature compensation(Figure 12b and Figure 13b). Moreover, the raw and actual fiber strain fluctuations after concrete cracks depend on the ambient temperature.

Meanwhile, the distributed expansive strain at the initial stage of specimen corrosion (about 0~40 h) after temperature compensation fail to be generated. The called corrosion rate threshold is about 1.34%, and cracking occurs at 160 h close to the observed cracking time of 155 h. The error is approximately 0.14%. Additionally, the distributed strain depends on a rising ambient temperature induced by the electro-accelerated corrosion.

To the end, the raw and actual strains from the distributed sensor monotonically increase over time and corrosion mass loss rate. Each curve shows three stages, being consistent with the degradation of reinforcement concrete. Regression analysis is performed to determine the equation of stages 2, 3 and 4. Before reinforcement concrete cracks, the slope of the fitting equation between the distributed strain and corrosion mass loss rate from the distributed sensor on the steel bar and mortar layer is 121 με/% and 156 με/%, as shown in Table 4 and Table 6, respectively. While their slopes of the fitting equation are 2.98 με/h and 3.41 με/h shown in Table 5 and Table 7, basically being consistent with 2-150# specimen. Simultaneously, the *P* values of regression equation are less than 0.05, thus the linear equation is credible.

To sum up, each curve can be divided into four stages (I, II, III, IV), which mirrors applied electrical current fluctuations and the degradation of concrete structure. A comparative analysis of specimens with protective layer thicknesses of 67 mm and 142 mm and an effective length of 200 mm shows that the corrosion rate thresholds in the first stages are related to the protective layer thickness of concrete structures. Furthermore, the crack corrosion rates observed the crack from the distributed strain are basically consistent with the actual corrosion rate optically observing crack.

Moreover, the corrosion sensitivity (stage II) before cracking of 2-150# and 2-300# concrete specimens is shown in Figure 11b, Figure 12b and Figure 13b, Table 2 and Table 6. The corrosion sensitivity of the distributed sensor located in steel bar is 125με/%. In the contrast, the corrosion sensitivity located in mortar layer is 156 με/%. These experimental sensitivities are in good agreement with the theoretical sensitivity of 92.8 με/%. The difference between the theoretical and measured data is attributed to these parameters such as corrosion expansion coefficient. Based on aforementioned corrosion sensitivity of 125, 156, 92.8 με/%, the tested error of a helix distributed sensor is lowered or enhance its the corrosion sensitivity further to attain a higher precision of measured corrosion mass loss.

## 6. Conclusions

To predict the corrosion rate of steel bar before the concrete crack, the authors put forward a prediction method using a spiral distributed optical fiber sensor. The main conclusions are as follows:(1) The mathematical model between steel corrosion rate and optical fiber is illustrated and the influencing factors of fiber strain are discussed here. The numerical analysis shows that the helical strain increases with the decrease of the thickness of concrete cover, declines with the increasing thickness of the porous zone, and grows with the corrosion expansion ratio.(2) In the process that reinforced concrete is corroded and broken, the strain measured from the spiral distributed optical fiber sensor is divided into four stages: (i) no strain stage; (ii) slow growth stage, (iii) abrupt change stage; and (iv) stabilized stage. These four stages are consistent with the concrete structure degradation: the corrosion products filling the porous region, the corrosion products accumulating until cracks generate, cracks propagating, and the equilibrium between corrosion products generation and outflow from cracks. (3) In the electro-accelerated corrosion test, a temperature fluctuation should be measured to eliminate the temperature error. The corrosion rate thresholds in the first stages depends on the protective layer thickness. In addition, the corrosive sensitivity of the proposed distributed sensor here is basically in a good agreement with a 92.8 με/% theoretical value. This work provides a feasible monitoring method and a practical prediction model for the corrosion loss rate.

## Figures and Tables

**Figure 1 micromachines-13-01868-f001:**
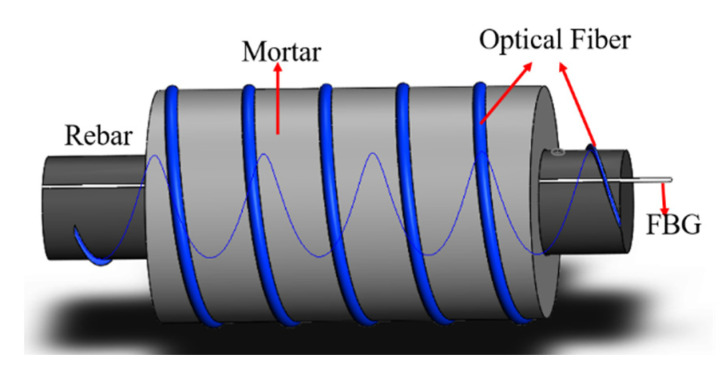
Basic structure of the spiral fiber sensor.

**Figure 2 micromachines-13-01868-f002:**
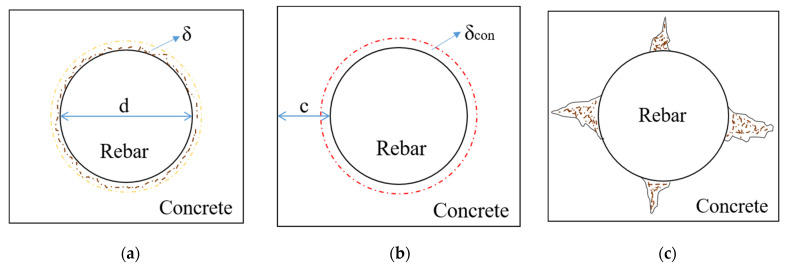
Corrosion process of the concrete. (**a**) First stage of corrosion; (**b**) Second stage of corrosion; (**c**) Third stage of corrosion.

**Figure 3 micromachines-13-01868-f003:**
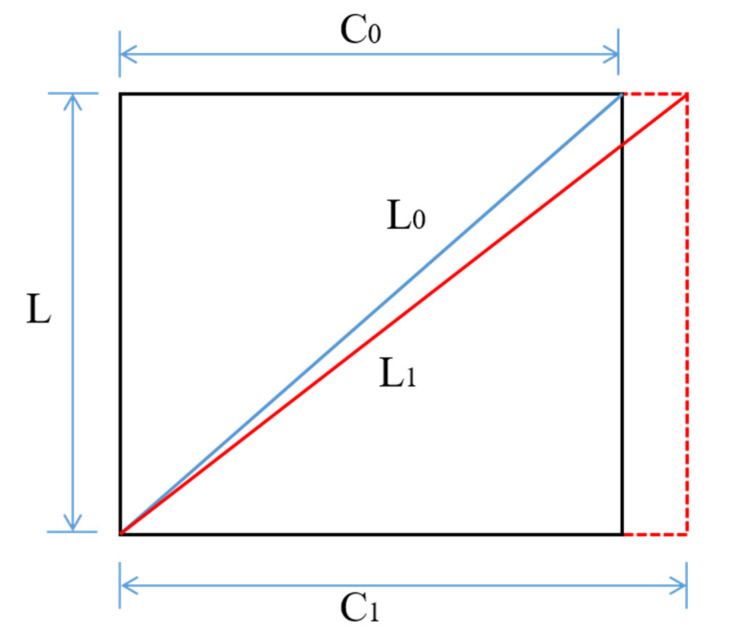
Two-dimensional sketch of unwind optical fiber.

**Figure 4 micromachines-13-01868-f004:**
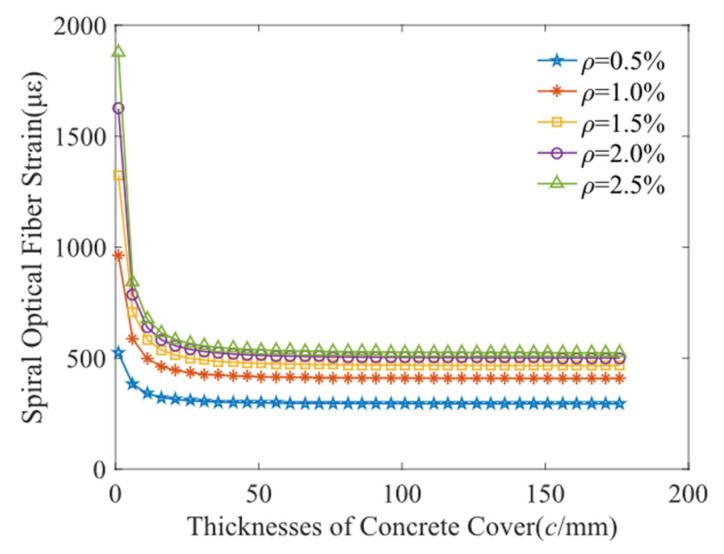
Fiber strain response with protective layer thickness.

**Figure 5 micromachines-13-01868-f005:**
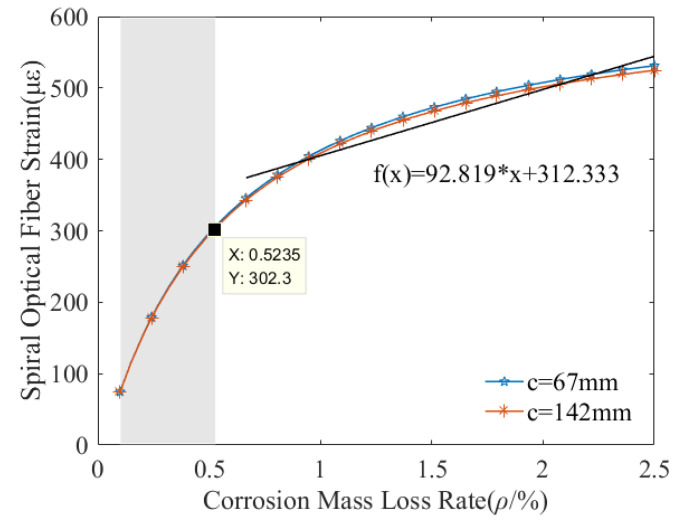
Fiber strain response to corrosion mass loss rate.

**Figure 6 micromachines-13-01868-f006:**
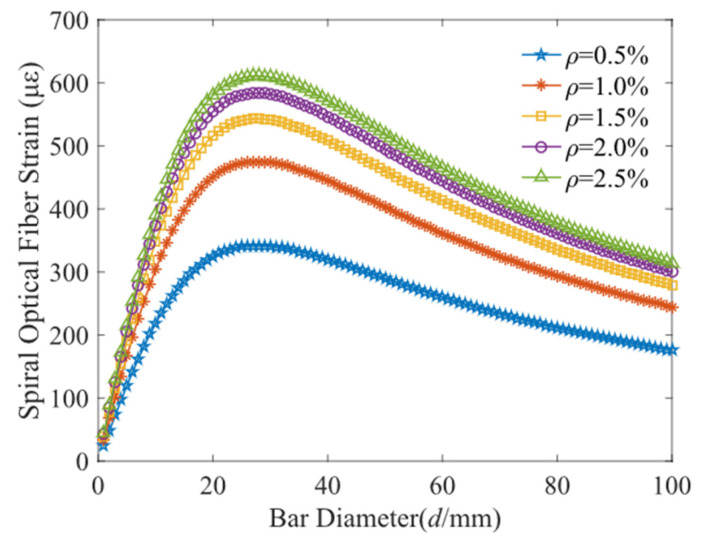
Fiber strain versus bar diameter.

**Figure 7 micromachines-13-01868-f007:**
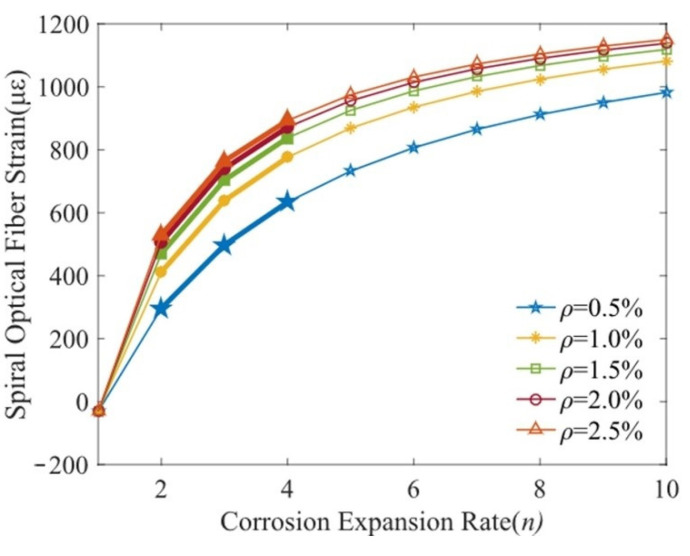
Fiber strain versus corrosion expansion coefficient.

**Figure 8 micromachines-13-01868-f008:**
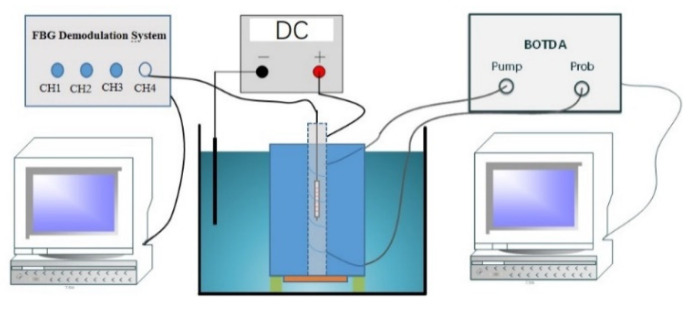
Schematic diagram of test device.

**Figure 9 micromachines-13-01868-f009:**
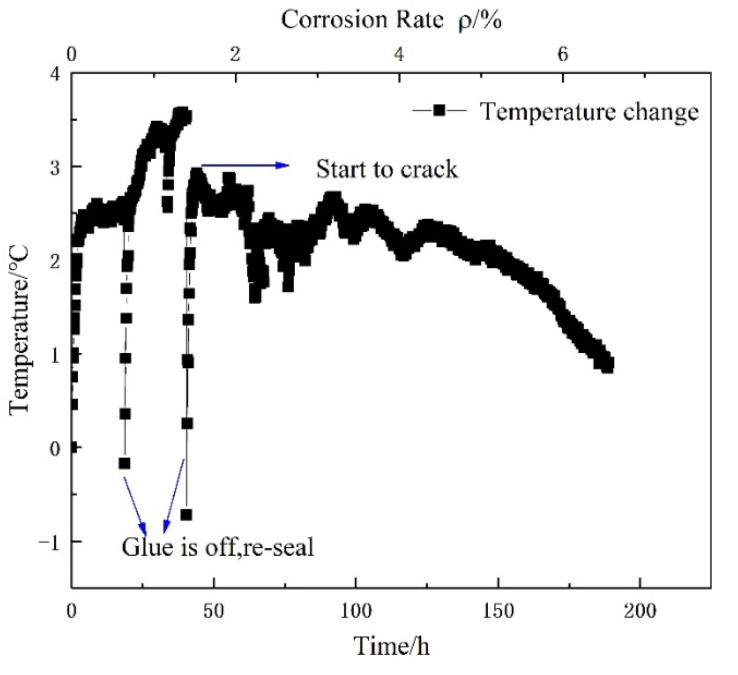
Temperature of rebar in electro-accelerated corrosion of 2-150# specimen.

**Figure 10 micromachines-13-01868-f010:**
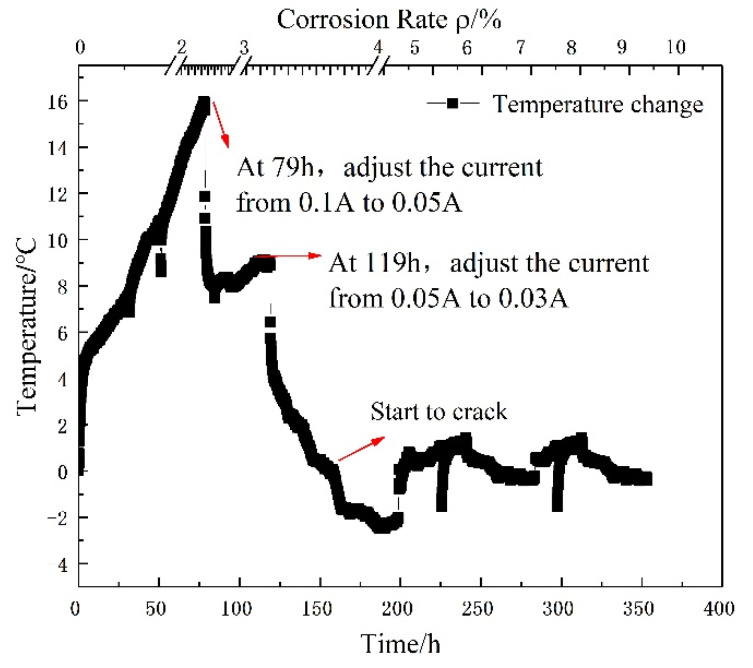
Temperature of rebar in electro-accelerated corrosion of 2-300# specimen.

**Figure 11 micromachines-13-01868-f011:**
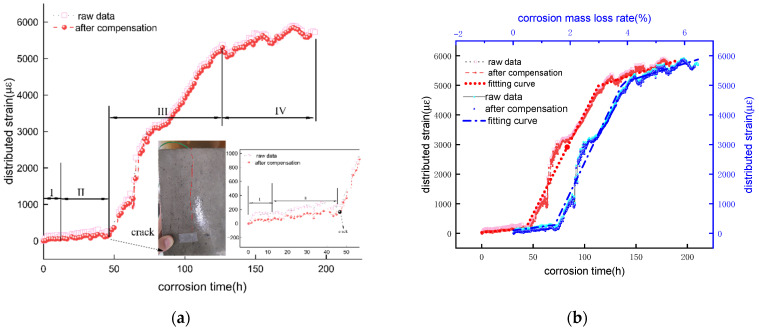
Fiber strain—time curve in mortar layer of 2-150# specimen. (**a**) raw data from reinforced layer distributed sensor; (**b**) fitting curve after temperature compensation.

**Figure 12 micromachines-13-01868-f012:**
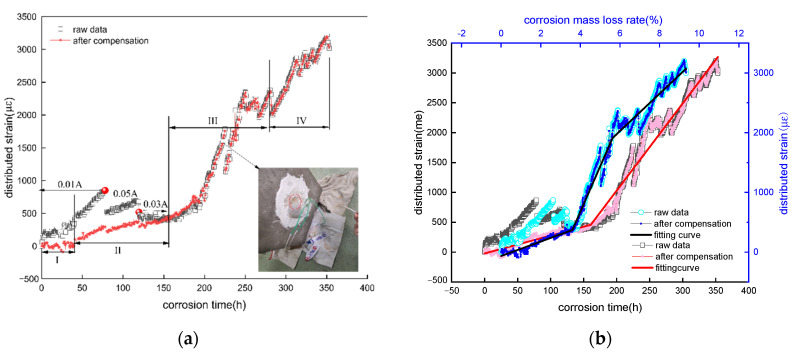
Fiber strain—time curve of reinforced layer of 2-300# specimen. (**a**) raw data from re-inforced layer distributed sensor; (**b**) fitting curve after temperature compensation.

**Figure 13 micromachines-13-01868-f013:**
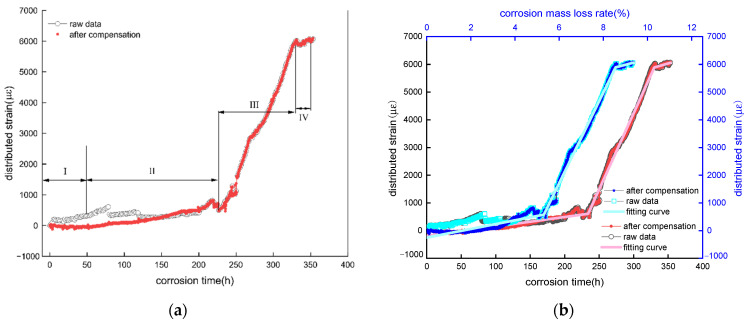
Fiber strain—time curve of mortar layer of 2-300# specimen. (**a**) raw data from reinforced layer distributed sensor; (**b**) fitting curve after temperature compensation.

**Table 1 micromachines-13-01868-t001:** Comparison with traditional corrosion monitoring methods.

	Optical Fiber Sensing	Half-Cell Potential	Electrochemical Impedance	Acoustic Emission	Electromagnetic Sensing
Non-Destructive Testing	Y	N	Y	Y	Y
Realtime measurement	Y	N	N	N	Y
Quantitative measurement	Y	N	N	N	N
anti-interference	Y	N	N	N	N

**Table 2 micromachines-13-01868-t002:** Test parameters for significance of fitting curve (red curve) of Figure 11b.

		Value	*t*-Value	Prob > |t|
Strain	A_1	98.30747	3.95197	8.23733 × 10^−5^
Strain	B_1	125.1493	4.22839	2.54594 × 10^−5^
Strain	B_2	2021.94744	140.84509	0
Strain	B_3	295.13916	24.4112	7.69873 × 10^−106^
Strain	A_2	−2671.61663		
Strain	A_3	3965.07549		

**Table 3 micromachines-13-01868-t003:** Test parameters for significance of fitting curve (blue curve) of Figure 11b.

		Value	*t*-Value	Prob > |t|
Strain	A_1	24.56684	0.97998	0.32731
Strain	B_1	3.37462	3.32703	9.06876 × 10^−4^
Strain	B_2	68.32792	138.76564	0
Strain	B_3	11.24715	25.91615	6.50915 × 10^−166^
Strain	A_2	−2769.68178		
Strain	A_3	3706.5015		

**Table 4 micromachines-13-01868-t004:** Test parameters for significance of fitting curve (blue curve) of Figure 12b.

	Value	*t*-Value	Prob > |t|
A_1	−64.07285	−8.21951	3.55932 × 10^−16^
Slope1	121.01677	38.41754	2.76929 × 10^−244^
Slope2	760.23007	105.48555	0
Slope3	312.77225	84.81807	0
A_2	−2365.37966		
A_3	151.20291		

**Table 5 micromachines-13-01868-t005:** Test parameters for significance of fitting curve (red curve) of Figure 12b.

	Value	*t*-Value	Prob > |t|
A_1	−21.01267	−2.3259	0.02012
Slope1	2.9852	31.15847	3.91241 × 10^−175^
Slope2	14.59532	200.9341	0
Slope3	11.57084	--	--
A_2	−1888.62174		
A_3	−674.59671		

**Table 6 micromachines-13-01868-t006:** Test parameters for significance of fitting curve (blue curve) of Figure 13b.

	Value	*t*-Value	Prob > |t|
A_1	−251.04553	−30.10543	9.8069 × 10^−166^
B_1	156.46386	60.62707	0
B_2	1635.02955	286.98289	0
B_3	210.0488	4.14511	3.53191 × 10^−5^
A_2	−8098.57618		
A_3	4095.15913		

**Table 7 micromachines-13-01868-t007:** Test parameters for significance of fitting curve (red curve) of Figure 13b.

	Value	*t*-Value	Prob > |t|
A_1	−188.89164	−33.08537	1.96872 × 10^−193^
B_1	3.41087	80.75018	0
B_2	55.05059	343.94794	0
B_3	7.1853	5.06991	4.33013 × 10^−7^
A_2	−12,274.16787		
A_3	3518.39254		

## Data Availability

The data presented in this study are available from the corresponding author, (J.-Z.L.), upon reasonable request.

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
