# Peer review of "Prediction Method of Steel Corrosion Rate Based on the Helix Distributed Sensor"

_micromachines, 2022, doi:10.3390/mi13111868_

Round 1

Reviewer 1 Report

This work proposes and demonstrates an optical fiber system to detect corrosion based on Fiber Bragg Grating (FBG). As the layer corrosion increases, the strain over the FBG increase, so it is possible to estimate the corrosion level. The authors propose and present a complete analysis considering essential parameters (temperature, strain, and time). In addition, the sensing setup and methods involve are novel. The manuscript is suitable for publication in Micromachines after minor comments. 

*The term "MATLAB analysis" is not suitable. The authors need to describe the analysis used. MatLab is not an analysis.  

*The authors must show the FBG  wavelength shifting analysis for the strain and temperature. Here, the authors need to include a discussion about cross-sensitivity. 

*A comparative table of the proposed technique and conventional methods to estimate corrosion will highlight the manuscript's contribution to the literature. Please describe the novelty of the work in terms of parameters to monitor.

Author Response

Dear Editor, Dear reviewers

Thank you very much for taking your time to review this manuscript. We truly appreciate all your comments and suggestions. Based on these comments, we have uploaded this letter for reviewer response. Accordingly, we uploaded the revised manuscript with change mode in Microsoft Office Word.

Appending this letter is our point-by-point response to the comments raised by the reviewers. Our responses are given directly afterwards for each comment.

We would also like to thank you for allowing us to resubmit a revised manuscript.

We hope that the revised manuscript is accepted for publication in the Micromachines.

*The term "MATLAB analysis" is not suitable. The authors need to describe the analysis used. MatLab is not an analysis.

Our response: We have revised this description and corrected it in the whole manuscript.

At line 15, "MATLAB analysis" revised to “numerical analysis by MATLAB”.

At line 75, "MATLAB analysis" revised to “numerical analysis with MATLAB”.

At line 472, "MATLAB analysis" revised to “numerical analysis”.

*The authors must show the FBG wavelength shifting analysis for the strain and temperature. Here, the authors need to include a discussion about cross-sensitivity.

Our response: the authors has added some text in this manuscript about the strain and temperature sensing characteristics of FBG. The added parts were as following,

“Due to a temperature change, the wavelength shift of FBG free from stress can be expressed as:

where  represents the thermal expansion coefficient of the fiber, which describes the relationship between the grating period and temperature;  represents the thermal optical coefficient of the fiber, which describes the relationship between the effective refractive index of the grating and temperature. Therefore, the temperature change can be obtained from the wavelength shift. The relationship between bare grating wavelength and temperature can be calculated as:

where k is 11 pm/℃. FBG located in the rubber sleeve is free in this experiment due to the much larger sleeve diameter than the diameter of FBG and is influenced by only temperature. “

Nevertheless, to present a comprehensive FBG theory in this manuscript, we have added this as a new section (section 2.2 in revised manuscript) to discuss the cross effect of temperature and stress.

*A comparative table of the proposed technique and conventional methods to estimate corrosion will highlight the manuscript's contribution to the literature. Please describe the novelty of the work in terms of parameters to monitor.

Our response:

We have added a table to compare optical fiber sensing with traditional methods in section 1 at of revised manuscript, the table as follow:

Optical fiber sensing

Half-cell potential

Electrochemical impedance

Acoustic emission

Electromagnetic sensing

Non-Destructive Testing

Y

N

N

N

Y

Continuous measurement

Y

N

Y

N

Y

Quantitative measurement

Y

N

Y

N

Y

anti-interference

Y

N

N

N

N

Thank you for the comments provided here. We believe we have addressed the comments in our response provided above. Your comments have helped to greatly improve our manuscript and are appreciated.

Reviewer 2 Report

The authors report in this work the development of a prediction method for steel corrosion mass loss rate measured by a spiral distributed fiber optic sensor. They present a reinforced concrete sample integrated with a spiral distributed fiber optic sensor and some experiment results where the helix strain along the distributed sensor was used to evaluate the corrosion mass loss of steel bar.

This paper is very interesting in my opinion but the document needs some major revisions regarding the measurements and sensor construction.

-Paragraph 2.2, the authors should add more information about the influence on the measure method due to the fibre shifting or pitch shifting during the corrosion phenomena.

- Line 189: The authors should correct the text …”Error! Reference source not found…”.

- In my opinion the authors should add some discussion about the dependence of measure method vs temperature changes , without the corrosion process.

- To better understand the novelty of this method the authors could be add a table that reports a comparison of different methods (at state of art) to measures the corrosion in such materials.

- the tables inside the figures 11, 12 (b) in my opinion should be moved outside the graphs.

- The authors should add some comments on limits (if any !!) on applying this method to a range of bar diameters.

Author Response

Dear Editor, Dear reviewers

Thank you very much for taking your time to review this manuscript. We truly appreciate all your comments and suggestions. Based on these comments, we have uploaded this letter for reviewer response. Accordingly, we uploaded the revised manuscript with change mode in Microsoft Office Word.

Appending this letter is our point-by-point response to the comments raised by the reviewers. Our responses are given directly afterwards for each comment.

We would also like to thank you for allowing us to resubmit a revised manuscript.

We hope that the revised manuscript is accepted for publication in the Micromachines.

-Paragraph 2.2, the authors should add more information about the influence on the measure method due to the fibre shifting or pitch shifting during the corrosion phenomena.

Our response: We have added some discussion in section 2.3 of revised manuscript to describe the influence of interval, as below:

“Meantime, a smaller interval will generate noise in the measurement results, and a larger interval will omit corrosion information which fails to accurately reflect the corrosion of reinforcement.”

Meantime, our manuscript has the sensor structural parameters such as an 8.7 cm and 6.5 cm winding pitch provided in these 136 and 137 lines. The fiber shifting or pitch shifting during corrosion is tiny, which does not cause to a light loss change and omit much more corrosion information compared with the initial pitch.

- Line 189: The authors should correct the text …”Error! Reference source not found…”.

Our response: A diagram is missing there, and we added it as Figure.7 in revised manuscript.

- In my opinion the authors should add some discussion about the dependence of measure method vs temperature changes, without the corrosion process.

Our response: We have added section 2.2 in revised manuscript which discuss the cross influence of stress and temperature. It shows that the temperature will not affect the monitoring results.

- To better understand the novelty of this method the authors could be add a table that reports a comparison of different methods (at state of art) to measures the corrosion in such materials.

Our response: We have added correlated text in this manuscript. to compare optical fiber sensing with traditional methods in section 1 at of revised manuscript, the table as follow:

Optical fiber sensing

Half-cell potential

Electrochemical impedance

Acoustic emission

Electromagnetic sensing

Non-Destructive Testing

Y

N

Y

Y

Y

Realtime measurement

Y

N

N

N

Y

Quantitative measurement

Y

N

N

N

N

anti-interference

Y

N

N

N

N

- the tables inside the figures 11, 12 (b) in my opinion should be moved outside the graphs.

  • Our response: We have revised these figures (Figure 11, Figure 12, Figure 13) and the tables inside the figures are listed in Table 2-Table7.

- The authors should add some comments on limits (if any !!) on applying this method to a range of bar diameters.

Our response: The following text was added in this manuscript.

“Based on aforementioned corrosion sensitivity of 125 ,156 ,92.8, the tested error of a helix distributed sensor is lowered or enhance its the corrosion sensitivity further to attain a higher precision of measured corrosion mass loss.

Thank you for the comments provided here. We believe we have addressed the comments in our response provided above. Your comments have helped to greatly improve our manuscript and are appreciated.

Round 2

Reviewer 2 Report

the paper is ready to be published.